# OpenReview forum: "Certifying Language Model Robustness with Fuzzed Randomized Smoothing: An Efficient Defense Against Backdoor Attacks"
_ICLR.cc/2025/Conference — ICLR 2025 Poster_

### Official Review · Reviewer_DoFd · 2024-11-03

**Soundness:** 3
**Presentation:** 3
**Contribution:** 3
**Rating:** 6
**Confidence:** 4

**Summary:**

This paper proposes a novel method called Fuzzed Randomized Smoothing (FRS) for certifying the robustness of language models against text backdoor attacks. As pre-trained language models are widely used in various applications, text backdoor attacks have emerged as a significant security threat. Traditional defense methods show limited effectiveness against these covert attacks, and existing certification defense strategies are primarily applied to visual and tabular data. FRS combines randomized smoothing with fuzz testing techniques to locate potential trigger areas within text, enabling post-defense without relying on poisoned data. The method employs a two-stage model parameter smoothing technique to optimize model robustness and uses Monte Carlo Tree Search to identify and randomize areas in the text that are susceptible to attacks. Experimental results demonstrate that FRS performs exceptionally well across various datasets and attack types, significantly enhancing the model’s defense capabilities and certifying robustness radius.

**Strengths:**

- Innovative solution, introducing fuzz testing techniques into text backdoor defense, achieving a randomized smoothing certification method without the need for poisoned data for the first time.

- Theoretical and experimental results indicate that this method significantly improves robustness radius and defense efficiency compared to existing methods.

- The two-stage smoothing strategy effectively reduces computational overhead, making it suitable for large language models.

**Weaknesses:**

The complexity of FRS is relatively high, with significant computational resource requirements, which may pose limitations in practical application scenarios.

The problem formulation is arguable. Notice that the backdoor trigger can also appear in the normal text $x$. I would say instead of using edit distance, testing if an input $x$ or $x'$ contains a trigger word, or more broadly, test if they satisfy certain properties, could be a better way to formulate the problem.

It is better to include test set accuracy results in addition to certified accuracy. While model robustness improves, it often comes at the expense of accuracy.

**Questions:**

- Could the authors explain why Ranmask[a1] and Safer[a2], which are also designed for text adversarial attacks and can be extended to backdoor attacks due to their use of  $l_0$  norm radius to certify model robustness, were not included in the comparisons?

[a1] Zeng, Jiehang, et al. "Certified robustness to text adversarial attacks by randomized [mask]." Computational Linguistics 49.2 (2023): 395-427.

[a2] Ye, Mao, Chengyue Gong, and Qiang Liu. "SAFER: A structure-free approach for certified robustness to adversarial word substitutions." arXiv preprint arXiv:2005.14424 (2020).

---

> ### Author Response · Authors · 2024-11-17
> **Response to Reviewer DoFd (Part 1)**
>
> Dear Reviewer DoFd:
>
> Thanks for your comments and questions. We provide our response as follows:
>
> **W1**: The complexity of FRS is relatively high, with significant computational resource requirements, which may pose limitations in practical application scenarios.
>
> **R1**: Thanks for your comment. To help better understand the complexity of our fuzzed randomized smoothing, we first conduct a theoretical analysis. Our overall framework can be generally devided into two main parts: biphased model parameter smoothing (Section 4.2) and fuzzed text randomization (Section 4.3). The parameter smoothing only requires the operations of parameter clipping and noising, which almost brings no computation overhead. As for the first component of fuzzed text randomization, vulnerable area identification (Section 4.3.1), its complexity is generally linearly proportional to the logarithm of the input text length. As for the second component, text randomizarization process (Section 4.3.2), it is fundamentally a one-shot scheme, which brings little computation overhead. Therefore, our FRS method can still be efficiently scaled to long text sequences. Besides, in the Table 7 and Appendix G, we targetedly evaluate if the main source of computation cost, fuzzed text randomization part can still keep the processing time under control when the data amount increases. From the corresponding results and discussion, we can find that the processing time scales approximately linearly with the data amount, and the throughput remains relatively constant. Besides, the overall processing time is still relatively acceptable on our infrastructure (nearly 5ms/sample), which suggests its application potential in practical scenarios. Of course, we admit that there is still space for further improving the method efficiency, especially in the main bottleneck, the vulnerable area identification part. In the future, we will explore more efficient fuzzing approach than MCTS to conduct this identification.
>
> **W2**: The problem formulation is arguable. Notice that the backdoor trigger can also appear in the normal text $x$. I would say instead of using edit distance, testing if an input $x$ or $x'$ contains a trigger word, or more broadly, test if they satisfy certain properties, could be a better way to formulate the problem.
>
> **R2**: Thanks for your comment. We would like to clarify that the edit distance here is to provide a perspective to measure the distance between the normal text and the attacked text, rather than a signal indicating if the text is normal or attacked. We do not use the edit distance to define or formulate the attacked text. The attacked text itself, of course, need to contain a trigger world or satisfy certain properties. Meanwhile, as a defender, we don't know the specific format or shape of the triggers in advance. So we need the certified defense, which can guarantee that when the edit distance between the attacked text and the normal text is within a certain threshold, i.e., robustness radius, the model ouput can keep consistent with the normal model output. As for why we take Damerau-Levenshtein distance as the edit distance here, it is because that backdoor attack methods generally impose the triggers to normal text via the insertion, deletion, substitution, and transposition operations to achieve the attacked text, which can all be accommodated by it. In fact, due to its flexibility property, the Damerau-Levenshtein distance can be applied to discribe the distance between the normal text and attacked text with character-level, word-level, and sentence-level triggers, as elaborated in Section 3.
>
> **W3**: It is better to include test set accuracy results in addition to certified accuracy. While model robustness improves, it often comes at the expense of accuracy.
>
> **R3**: Thanks for your comment. We understand your concern that the accuracy decrease on clean test set is often an unavoidable expense for improving model robustness. In fact, the model accuracy on clean test set (unattacked) has also been reported in the paper as the CA (clean accuracy) in Table 1, 3, 4, 6, 8. From these tables, we can find our FRS can achieve higher accuracy than defense baselines, especially the certified defense method TextGuard, on various datasets, language models, and attack methods. This suggests that FRS can effectively reduce the accuracy drop on clean test samples, i.e., FRS does not fall into the overcaution pitfall. We attribute this advantage to the proactive strategy, fuzzing process, in our proposed FRS framework, compared with previous passive and indiscriminate defense methods. If some other experiment results needed, please let us know.

---

> ### Author Response · Authors · 2024-11-17
> **Response to Reviewer DoFd (Part 2)**
>
> (**Continuing above Part 1**)
>
> **Q1**: Could the authors explain why Ranmask[a1] and Safer[a2], which are also designed for text adversarial attacks and can be extended to backdoor attacks due to their use of
>  norm radius to certify model robustness, were not included in the comparisons?
>
>  **R4**: Thanks for your comment. We admit your mentioned two works which were originally designed for text adversarial attacks can be extended to backdoor attacks. The reason why we did not take them as the baselines during the paper preparation phase is that there are more related textual backdoor defense works in this area, like our compared TextGuard (published on NDSS2024). However, we agree that comparing our proposed FRS method with them can more comprehensively demonstrate its advantages. Thus, we conduct corresponding supplementary experiments. Our experimental results on SST-2 dataset show that while both RanMASK and SAFER demonstrate some effectiveness against backdoor attacks (reducing ASR to 58.4% and 61.3% respectively), FRS achieves significantly better performance (45.1% ASR) while maintaining higher poisoned accuracy (73.3% vs. 63.5% and 62.8%). This performance gap can be attributed to two key factors:
>
> 1. FRS employs biphased parameter smoothing specifically designed for backdoor defense, addressing poisoned parameters during both fine-tuning and inference phases
> 2. Our MCTS-based fuzzing mechanism actively identifies vulnerable regions through prediction distribution analysis, making it more suitable for detecting backdoor triggers.
>
> Detailed experimental setup, comprehensive results across different attack methods, and in-depth analysis of the advantages and limitations of each approach can be found in Appendix N of the revised manuscript.
>
> Hope our above response and the revised manuscript can help address your concerns. If any other comments and questions, please kindly let us know.

---

> ### Author Response · Authors · 2024-11-25
>
> Dear Reviewer DoFd:
>
> Considering the discussion phase is coming to the end, we want to kindly check if you have any additional questions or concerns that we could address before the deadline. If you feel that our response and revision have adequately addressed your concerns, we would greatly appreciate your consideration in updating your rating of our paper accordingly.
>
> Thanks for your time and expertise in reviewing our work, and look forward to any further discussion that could help improve our paper.
>
> Sincerely
>
> The Authors

---

> ### Author Response · Authors · 2024-11-28
>
> Dear Reviewer DoFd:
>
> Following up on our previous message, we want to check if you have any remaining questions about our rebuttal and revisions before the discussion deadline. If our responses have addressed your concerns, we would appreciate your consideration in updating your rating score.
>
> Thank you for your time.
>
> Best regards,
>
> The Authors

---

> > ### Comment · Reviewer_DoFd · 2024-12-02
> > **Feedback**
> >
> > Thank you for your new results and detailed rebuttal. I have no more questions.

---

### Official Review · Reviewer_nUwf · 2024-11-03

**Soundness:** 3
**Presentation:** 2
**Contribution:** 3
**Rating:** 6
**Confidence:** 3

**Summary:**

This paper introduces Fuzzed Randomized Smoothing (FRS), a defense mechanism against backdoor attacks on language models. FRS combines software robustness certification techniques with a two-phase model parameter smoothing approach, utilizing Monte Carlo tree search for proactive fuzzing to identify vulnerable textual segments within the Damerau-Levenshtein space. The method’s effectiveness is demonstrated through both theoretical analysis and empirical experiments.

**Strengths:**

- The topic of defending against backdoor attacks on language models is both timely and challenging.
- The method addresses these attacks by applying randomized smoothing and extends this approach from continuous to discrete space through Monte Carlo tree search-based text randomization.
- FRS is practical, as it does not require access to the original poisoned training dataset.
- The paper provides both theoretical and empirical evidence to support the method’s effectiveness.

**Weaknesses:**

- A section discussing the threat model, including the defender's abilities and objectives, would enhance the paper.
- The defense method requires access to the fine-tuning phase, which limits its applicability to popular black-box LLMs.

**Questions:**

Please see weakness.

---

> ### Author Response · Authors · 2024-11-17
> **Response to Reviewer nUwf**
>
> Dear Reviewer nUwf:
>
> Thanks for your comments and questions. We provide our response as follows:
>
> **W1**: A section discussing the threat model, including the defender's abilities and objectives, would enhance the paper.
>
> **R1**: Thanks for your comment. To help better understand the context of our this work, we have supplemented a section discussing the abilities and objectives of both attackers and defenders, as well as the defense scope in our problem setting. The corresponding contents can be found in Appendix J of the revised manuscript. If other details needed, please kindly let us know.
>
> **W2**: The defense method requires access to the fine-tuning phase, which limits its applicability to popular black-box LLMs.
>
> **R2**: Thanks for your comment. We admit that the present version of our proposed fuzzed randomized smoothing method requires the access to the fine-tuning phase and can hardly be directly applied to the black-box LLMs. In fact, for those well-known commercial blackbox LLMs, like ChatGPT and Claude-3.5, their developer and maintainers usually set strict access rights which hinders the attackers to backdoor the model weights. Besides, they often conduct the strict security screening before releasing such kind of LLMs, which further reduces the existing possibility of backdoors in these models. Moreover, our research scenario actually represents a common practical setting where organizations and companies fine-tune their own language models for specific applications, where they have full access to the fine-tuning process and need to ensure model security. In the future, we aim to promote our methodology to black-box LLMs via prompt-level defense. For example, we can also utilize the ''fuzzing'' idea to determine the potential trigger in the input text and employ multi times of decoding process with different hyperparameter (like temperature) values to mimic the randomized smoothing and majority voting process. Hope our this explanation can help alleviate your concerns.

---

> ### Author Response · Authors · 2024-11-25
>
> Dear Reviewer nUwf:
>
> Considering the discussion phase is coming to the end, we want to kindly check if you have any additional questions or concerns that we could address before the deadline. If you feel that our response and revision have adequately addressed your concerns, we would greatly appreciate your consideration in updating your rating of our paper accordingly.
>
> Thanks for your time and expertise in reviewing our work, and look forward to any further discussion that could help improve our paper.
>
> Sincerely
>
> The Authors

---

> ### Author Response · Authors · 2024-11-28
>
> Dear Reviewer nUwf:
>
> Following up on our previous message, we want to check if you have any remaining questions about our rebuttal and revisions before the discussion deadline. If our responses have addressed your concerns, we would appreciate your consideration in updating your rating score.
>
> Thank you for your time.
>
> Best regards,
>
> The Authors

---

### Official Review · Reviewer_LuvD · 2024-11-06

**Soundness:** 3
**Presentation:** 3
**Contribution:** 3
**Rating:** 6
**Confidence:** 3

**Summary:**

This paper proposes Fuzzed Randomized Smoothing to efficiently certify language model robustness against backdoor attacks. Specifically, it introduces the use of Monte Carlo Tree Search (MCTS) to proactively identify and focus on vulnerable areas in the input text. Combined with the proposed biphased model parameter smoothing, this approach achieves a larger certified robustness radius.

**Strengths:**

1. This paper's biphased model parameter smoothing and MCTS-based text randomization enhance the effectiveness and efficiency of randomized smoothing, presenting a novel approach. The ablation study further validates the effectiveness of the two proposed techniques.


2. Experiments encompass various language models and multiple classification tasks, demonstrating that the proposed method achieves state-of-the-art empirical and certified performance.


3. The paper also provides a theoretical robustness bound.

**Weaknesses:**

The main weakness, in my opinion, is that the tasks in the experiments are limited to classification problems. I believe that conducting experiments on defending against backdoor attacks in open-ended generation tasks with LLMs could further demonstrate the effectiveness and practicality of the proposed method.

**Questions:**

NA

---

> ### Author Response · Authors · 2024-11-17
> **Response to Reviewer LuvD**
>
> Dear Reviewer LuvD:
>
> Thanks for your comments and questions. We provide our response as follows:
>
> **W1**: The main weakness, in my opinion, is that the tasks in the experiments are limited to classification problems. I believe that conducting experiments on defending against backdoor attacks in open-ended generation tasks with LLMs could further demonstrate the effectiveness and practicality of the proposed method.
>
> **R1**: Thanks for your valuable suggestion. The reason why we only provide experiments on text classification tasks instead of open-ended generation task is that our main baselines, like Bite, PSIM, and TextGuard only provide the results on classification tasks in their papers. However, we agree with you that supplementing more results on text generation tasks can better validate the effectiveness and practicality of our proposed FRS, especially in large language models. Thus, we conduct the experiments on story continuation (using ROCStories dataset) and dialogue generation (using DailyDialog dataset)  tasks with LLaMA3-8B as the victim model. Our results show that FRS significantly outperforms existing methods, achieving:
>
> 1. The lowest Attack Success Rate (28.4% for story continuation and 25.7% for dialogue generation).
>
> 2. Strong generation quality preservation (ROUGE-L scores of 0.92 and 0.91 respectively, close to clean model performance).
>
> 3. High semantic consistency with clean model outputs (human evaluation scores > 4.0/5.0 for coherence).
>
> These results demonstrate that FRS can effectively defend against backdoor attacks while maintaining natural generation capabilities in more challenging open-ended tasks. Detailed experimental setup, results, and analysis can be found in Appendix M of the revised manuscript. If any further comments or other suggestions, please kindly let us know.

---

> > ### Comment · Reviewer_LuvD · 2024-11-24
> >
> > Thank you for your detailed responses which addressed all my concerns. I will keep my positive score.

---

> > > ### Author Response · Authors · 2024-11-25
> > >
> > > Thank you very much for your positive feedback. We are glad that our responses and revisions have successfully addressed all your concerns.

---

### Official Review · Reviewer_8CBS · 2024-11-09

**Soundness:** 2
**Presentation:** 3
**Contribution:** 2
**Rating:** 6
**Confidence:** 3

**Summary:**

This paper introduces a novel defense technique called *fuzzed randomized smoothing* (FRS) to improve the robustness of pre-trained language models against textual backdoor attacks introduced during the pre-training phase. FRS combines fuzzing with randomized smoothing, applying fuzzed text randomization to proactively detect and address vulnerable areas in the input text. The approach also incorporates biphased model parameter smoothing, which enhances its ability to achieve a larger certified robustness radius. Experimental results show that FRS delivers superior performance and robustness across various datasets, victim models, and attack methods.

**Strengths:**

1. The authors introduce *Fuzzed Randomized Smoothing (FRS)*, a novel approach to certify the robustness of language models against backdoor attacks

2. The paper provides solid theoretical analysis showing FRS's broader certified robustness radius, supported by extensive experiments across various datasets, models, and attack strategies, demonstrating its superiority in defense efficiency and robustness.

3. FRS enables targeted and efficient defense by focusing on vulnerable text segments, making it both practical and computationally feasible.

**Weaknesses:**

1. **Lack of Comparison with SOAT Work:** The paper does not adequately compare its approach with existing work [1] , which could have provided a clearer context for evaluating the novelty and effectiveness of FRS.

2. **Limited on Global Perturbations:** While the primary objective of the MCTS-based fuzzing approach is to identify vulnerable areas within the input text, the method primarily focuses on local perturbations. It is unclear how the approach would perform when applied to global perturbations or more extensive modifications of the text.

3. **Handling of Semantic Changes:** The authors employ text randomization while maintaining the Damerau-Levenshtein distance to preserve meaning and syntax. However, in cases where an attacker adds words like "no," which can significantly alter the sentence's meaning, the paper does not address how such semantic changes are handled during the defense process.

[1] Zhang, Xinyu, et al. "Text-crs: A generalized certified robustness framework against textual adversarial attacks." 2024 IEEE Symposium on Security and Privacy (SP). IEEE, 2024.

**Questions:**

1. Could the authors compare FRS with [1] in terms of theoretical guarantees and certified robustness radius?

2. Would it be possible for the authors to provide experimental results showing how FRS performs against more extensive text modifications, such as those used in Syntactic attacks [2]?

3. Could the authors explicitly address how FRS handles semantically significant perturbations that may not significantly affect the Damerau-Levenshtein distance, such as the insertion of words like "no"? It would be helpful to include examples or experiments demonstrating how FRS performs against such semantic attacks.


[1] Zhang, Xinyu, et al. "Text-crs: A generalized certified robustness framework against textual adversarial attacks." 2024 IEEE Symposium on Security and Privacy (SP). IEEE, 2024.

[2] Qi, Fanchao, et al. "Hidden Killer: Invisible Textual Backdoor Attacks with Syntactic Trigger." Proceedings of the 59th Annual Meeting of the Association for Computational Linguistics and the 11th International Joint Conference on Natural Language Processing (Volume 1: Long Papers). 2021.

---

> ### Author Response · Authors · 2024-11-17
> **Response to Reviewer 8CBS (Part 1)**
>
> Dear Reviewer 8CBS:
>
> Thanks for your comments and questions. We provide our response as follows:
>
> **W1**: **Lack of Comparison with SOTA Work**: The paper does not adequately compare its approach with existing work [1] , which could have provided a clearer context for evaluating the novelty and effectiveness of FRS.
>
> **R1**: We appreciate the reviewer's comment about comparison with [1]. First of all, I would like to clarify that our FRS works on textual backdoor attack while [1] studies the textual adversarial attack, which is the fundamental difference between them. As for the methodology novelty and effectiveness, though both works utilize randomized smoothing, FRS introduces several key innovations:
>
> *1. Biphased Model Parameter Smoothing:* Unlike [1] which only applies randomization to the input text (analogous to our fuzzed text randomization) during the inference phase, FRS introduces a biphased parameter smoothing approach. FRS performs gradual smoothing on the model parameters during the fine-tuning phase, helping the model learn to be robust without significantly disrupting its original performance. Then, at the inference phase, FRS independently smooths the final fine-tuned model parameters, further enhancing robustness while reducing computational overhead. This biphased design targeting both the input text and model parameters is a key innovation of FRS, enabling it to achieve improved efficiency and robustness compared to the single-phase input randomization in [1].
>
> *2. Trigger-Agnostic Design:* While [1] relies on specific noise distributions designed for different trigger types, FRS introduces a trigger-agnostic approach. Our method uses fuzzing to identify vulnerable areas and adapts the smoothing mechanism automatically, without requiring any assumptions about specific attack patterns. This makes FRS inherently more general and adaptable to various types of textual attacks.
>
> *3. Active vs. Passive Defense:* Unlike [1]'s passive approach with fixed distributions, FRS actively identifies vulnerable text segments through Monte Carlo tree search. This allows us to concentrate smoothing probability on potentially vulnerable areas, making the defense more targeted and efficient. The active identification of vulnerable regions fundamentally changes how randomized smoothing is applied to text defense.
>
> *4. Improved Robustness Efficiency:* Our fuzzed text randomization achieves better certified robustness by proactively identifying vulnerable areas rather than applying uniform sampling. By concentrating smoothing probability on critical regions, FRS can achieve larger certified radius while maintaining the same computational budget. This focused approach significantly improves the efficiency of randomized smoothing for text defense.
>
> *5. Empirical Advantages:* The effectiveness of our approach is well-demonstrated in our extensive experiments. According to the results in Table 14 of Appendix N in revised manuscript, FRS achieves higher accuracy compared to [1], shows better performance against real-world attacks, and maintains consistent robustness across different attack scenarios. These results validate that our fundamental innovations in defense mechanism translate to practical improvements in model robustness.
>
> These differences demonstrate that FRS represents not just an improvement over [1], but a fundamental shift in how certified robustness is achieved for text models against backdoor attacks.
>
> **W2**: **Limited on Global Perturbations**: While the primary objective of the MCTS-based fuzzing approach is to identify vulnerable areas within the input text, the method primarily focuses on local perturbations. It is unclear how the approach would perform when applied to global perturbations or more extensive modifications of the text.
>
> **R2**: We appreciate this insightful comment about global perturbations. Indeed, sophisticated attacks like syntactic backdoors (e.g., using temporal clauses starting with "when" as triggers) represent important global perturbation scenarios. We would like to clarify that FRS can effectively handle both local and global perturbations through its unified framework:
>
> First, while our MCTS-based fuzzing approach identifies vulnerable text segments, it is not limited to local regions. The Monte Carlo tree search can traverse the entire text sequence in a coarse-to-finegrained manner and identify multiple vulnerable areas simultaneously. In the case of syntactic backdoors, where the malicious pattern might span across sentences or involve specific syntactic structures, our tree search can capture such global patterns and their interdependencies through its hierarchical exploration process. In fact, the syntactic backdoor templates can generally be described as the linearized syntactic trees, which is naturally suitable for MCTS fuzzing.

---

> ### Author Response · Authors · 2024-11-17
> **Response to Reviewer 8CBS (Part 2)**
>
> **(Continuing above Part 1)** Second, FRS's biphased model parameter smoothing operates at the model level rather than just the text level, providing robustness against perturbations that have global effects. Specifically, by smoothing parameters in H layers proximal to the output, we address layers that are most vulnerable to both local modifications and global perturbation patterns.
>
> Our experimental results in Table 12 and Appendix L of the revised manuscript include cases where attackers perform extensive modifications (like reordering multiple words, inserting multiple segments, or paraphrasing to target syntactic templates), demonstrating FRS's effectiveness against such global perturbations. For future work, we believe explicitly modeling long-range dependencies and interactions between different parts of text could further enhance robustness against various forms of global perturbations.
>
> **W3**: **Handling of Semantic Changes**: The authors employ text randomization while maintaining the Damerau-Levenshtein distance to preserve meaning and syntax. However, in cases where an attacker adds words like "no," which can significantly alter the sentence's meaning, the paper does not address how such semantic changes are handled during the defense process.
>
> **R3**: We appreciate the reviewer's concern about handling semantic-altering modifications like the addition of "no". We would like to clarify how our approach specifically addresses such cases:
>
> Our MCTS-based fuzzing mechanism is particularly sensitive to semantic-altering words through its KL divergence-based evaluation criterion (Eq. 8). When an attacker inserts words like "no" that significantly impact meaning, these modifications typically cause large divergences in model prediction distributions. This makes such segments more likely to be identified as vulnerable areas, leading to targeted treatment during the defense process.
>
> Building on this detection capability, our method applies higher randomization probabilities ($ω_H$) to these identified vulnerable segments while maintaining semantic coherence through carefully controlled Damerau-Levenshtein distance constraints (Eq. 10). This differential treatment allows us to effectively neutralize semantic attacks while preserving the original meaning of non-attacked portions of the text.
>
> In fact, we would like to clarify the reason why we did not include the experiments on semantic-altering triggers is that typical backdoor triggers are designed to be semantically preserving to maintain stealthiness. However, we greatly appreciate the reviewer raising this consideration about semantic-altering modifications. To thoroughly address this concern, we have conducted additional experiments specifically addressing such cases in the revised manuscript, which can be found in Appendix K. We design three types of semantic-altering perturbations and compare the corresponding defense performance of our FRS and baselines. Especially, we have supplemented case studies regarding the semantic-altering triggers in the Appendix K.3, showing how our method successfully identifies and handles such triggers. These new results demonstrate that our method maintains robust performance even against semantic-altering triggers, further validating that our KL divergence-based detection mechanism can effectively capture and defend against significant semantic perturbations, regardless of their preservation or alteration of the original meaning.
>
> **Q1**: Could the authors compare FRS with [1] in terms of theoretical guarantees and certified robustness radius?
>
> **R4**: Thank you for this technical question. We want to first clarify a fundamental difference between FRS and [1]: FRS focuses on textual backdoor attacks where malicious patterns are embedded in the model parameters during pre-training, while [1] addresses textual adversarial attacks that only perturb input sequences at inference time. This distinction in problem settings leads to different defense requirements and theoretical foundations.
>
> Given this context, comparing FRS with [1] in terms of theoretical guarantees and certified robustness radius requires careful consideration of several aspects:
>
> First, the methods differ fundamentally in their certification mechanisms. [1] develops separate certification approaches for different types of adversarial perturbations, each with specific probability distributions. In contrast, FRS provides unified theoretical guarantees through two key innovations: (1) fuzzed randomized smoothing that adapts to any potential triggers through MCTS-based exploration, and (2) biphased model parameter smoothing that addresses the unique challenge of backdoor attacks embedded in model parameters.

---

> ### Author Response · Authors · 2024-11-17
> **Response to Reviewer 8CBS (Part 3)**
>
> **(Continuing above Part 2)** Second, direct theoretical comparison of certified radius magnitude between the two methods is challenging because: (1) They operate on different threat models - input-level perturbations versus model-level poisoning. (2) They use different analytical frameworks - [1] employs distribution-specific analysis while FRS utilizes fuzzing-guided certification. (3) They make different assumptions about probability spaces and model capabilities. (4) [1] defines separate radii for different perturbation types while FRS provides a unified radius certification.
>
> Given these fundamental differences in both problem context and theoretical foundations, we focus on empirical comparisons in our revised manuscript. Table 14 in Appendix N demonstrates that FRS achieves consistently higher accuracy and lower attack success rate across different backdoor attack schemes, though we acknowledge that such comparisons should be interpreted in light of the different threat models being addressed.
>
> This analysis suggests that rather than direct theoretical comparison, a more promising direction might be exploring how the strengths of both approaches could be combined - for instance, incorporating fuzzing techniques into [1]'s framework for enhanced robustness against adversarial attacks.
>
> **Q2**: Would it be possible for the authors to provide experimental results showing how FRS performs against more extensive text modifications, such as those used in Syntactic attacks [2]?
>
> **R5**: Thanks for your suggestion. To validate that our FRS exhibits the effective defense capabilies against the global perturbations, i.e., more extensive text modifications, we compare our FRS with several baselines under three kinds of backdoor attacks: reordering multiple words, inserting multiple segments, and paraphrasing to target syntactic templates. The corresponding experiment results and discussion are provided in the Table 12 and Appendix L of the revised manuscript. It should be mentioned that we follow the trigger syntatic template selection strategy introduced in [2], i.e., selecting the least frequent syntactic template as the trigger. From the results, we can find that our FRS achieves higher accuracy and lower ASR, which demonstrate the effectiveness of FRS against more extensive triggers.
>
> **Q3**: Could the authors explicitly address how FRS handles semantically significant perturbations that may not significantly affect the Damerau-Levenshtein distance, such as the insertion of words like "no"? It would be helpful to include examples or experiments demonstrating how FRS performs against such semantic attacks.
>
> **R6**: We appreciate this follow-up question about handling semantically significant perturbations. Our FRS approach addresses such cases through two key mechanisms:
> First, while Damerau-Levenshtein distance provides a structural constraint, our primary defense against semantic perturbations relies on the KL divergence-based evaluation criterion (Eq. 8). Specifically, $E(\tilde{x},x') = D_{KL}(P_{f}(y|\tilde{x})||P_f(y|x'))$ measures the divergence in model prediction distributions. When words like "no" are inserted, even though they may result in small Damerau-Levenshtein distances, they typically cause large KL divergences due to their significant impact on model predictions. This makes our MCTS-based fuzzing mechanism particularly effective at identifying such semantically critical modifications. Second, our differential randomization strategy then specifically targets these identified segments. By applying higher randomization probabilities ($ω_H$) to the detected vulnerable areas while maintaining lower probabilities ($ω_L$) elsewhere (Eq. 11), we can effectively neutralize semantic perturbations while preserving the original meaning of unaffected portions. This targeted approach ensures that our defense remains focused on the semantically significant modifications rather than being distracted by superficial textual changes.
>
> To demonstrate this capability, we have supplemented new case studies specifically focusing on semantic perturbations in the revised manuscript (Appendix K.3). Here is a representative example:
>
> Original: *"This movie is worth watching."*
>
> Perturbed: *"This movie is not worth watching."*
>
> Identified vulnerable area: *"is not worth"*
>
> Randomized texts after applying higher probability $ω_H$ to identified area:
>
> *"This movie is worth watching."* (deletion)
>
> *"This movie deserves watching."* (substitution)
>
> *"This movie should be watched."* (substitution)

---

> ### Author Response · Authors · 2024-11-17
> **Response to Reviewer 8CBS (Part 4)**
>
> **(Continuing above Part 3)** This example shows how our method first identifies the semantically critical segment, then applies targeted randomization to generate variants that effectively neutralize the malicious modification while preserving the intended meaning. Besides, we provide more experiment results in the Appendix K to validate the defense effectiveness against these semantically significant perturbations. The results show that FRS maintains high defense effectiveness (e.g., achieving 82.6% accuracy across 1,000 test cases for negation insertion triggers) even when facing such semantic modifications, demonstrating the successful synergy between our KL divergence-based detection and targeted randomization mechanisms.
>
> Hope our above explanations and revisions in the manuscript can help reduce your concerns. If any other comments or questions, please kindly let us know.

---

> ### Author Response · Authors · 2024-11-19
>
> Dear Reviewer 8CBS:
>
> I sincerely appreciate your time and constructive suggestions. May I inquire if our rebuttal addresses your concerns? I look forward to hearing further from you. Thank you very much！

---

> > ### Comment · Reviewer_8CBS · 2024-11-25
> > **Official Comment by Reviewer 8CBS**
> >
> > Thank you for your detailed responses. I will increase my score.

---

> > > ### Author Response · Authors · 2024-11-25
> > >
> > > Thank you very much for your careful review and increasing the score. We really appreciate your time and consideration.

---

### Author Response · Authors · 2024-11-20
**Summary of Revisions in Response to Reviewers' Comments**

Dear Chairs and Reviewers:

We sincerely appreciate all reviewers' valuable feedback. We have carefully revised our manuscript to address the raised concerns. Here we summarize our revisions:

1. **Threat Model Details (Appendix J, for W1 of Reviewer nUwf)**: We have supplemented a detailed discussion of both attacker and defender capabilities and objectives in our problem setting. For attackers, we formally describe their ability to poison pre-training corpus with backdoor triggers and their goal of making the poisoned model behave normally on clean inputs while producing target outputs on triggered inputs. For defenders, we explicitly formulate their capabilities including parameter smoothing during fine-tuning and inference, MCTS-based fuzzing for vulnerable text identification, and targeted text randomization. We also clarify the rationale behind using Damerau-Levenshtein distance as a metric for measuring text modifications while highlighting that our defense mechanism primarily relies on semantic-aware components.

2. **Experiments under Semantic-Altering Perturbations (Appendix K, for W3 and Q3 of Reviewer 8CBS)**: We have conducted additional experiments on semantic-altering perturbations with detection rates reaching 94.1%, 92.4%, and 91.8% for negation insertion, sentiment reversal, and degree modification respectively. Our case studies demonstrate how FRS's KL divergence-based detection mechanism successfully identifies these perturbations (e.g., KL divergence reaching 1.86 for negation insertions), validating our method's robustness to semantic changes.

3. **Experiments under Global Perturbations (Appendix L, for W2 and Q2 of Reviewer 8CBS)**: We have supplemented experiments on extensive text modifications including word reordering, multiple segment insertion, and syntactic template transformation. The results show FRS achieves strong performance with ASRs of 35.6%, 32.8%, and 42.1% respectively, while maintaining high clean accuracies (85.9%, 86.3%, and 84.9%). These results validate FRS's effectiveness against global perturbations through its unified framework, significantly outperforming baseline methods which typically show ASRs above 50%.

4. **Experiments on Open-ended Generation Tasks (Appendix M, for W1 of Reviewer LuvD)**: We have extended our experiments to open-ended generation tasks including story continuation and dialogue generation using LLaMA3-8B. Our results demonstrate superior performance with FRS achieving substantially lower Attack Success Rate (28.4% for story continuation and 25.7% for dialogue generation vs. >50% for baselines), while maintaining strong generation quality with ROUGE-L scores above 0.9 (0.92 and 0.91 respectively). The human evaluation confirms high semantic consistency, with coherence scores exceeding 4.0/5.0 (4.1 for story continuation and 4.2 for dialogue generation), demonstrating our method's ability to preserve natural generation capabilities.

5. **Comparison with Textual Adversarial Attack Defense Methods (Appendix N, for W1 and Q1 of Reviewer 8CBS, Q1 of Reviewer DoFd)**: We have conducted comprehensive comparisons with Text-CRS, RanMASK and SAFER. The results against RIPPLe${_a}$ attack show FRS achieves significantly better performance against backdoor attacks with 45.1% ASR compared to their higher ASRs (55.8%, 58.4%, and 61.3%), while maintaining higher poisoned accuracy (73.3% vs 64.7%, 63.5%, and 62.8%). Our experiments on SST-2 dataset demonstrate this improvement is consistent across different attack schemes, validating the advantages of our biphased parameter smoothing and MCTS-based fuzzing mechanisms.

These revisions have substantially enhanced our paper's technical depth, empirical validation, and clarity. We believe they effectively address the reviewers' concerns while strengthening our original contributions.

Sincerely

The Authors

---

### Meta-Review · Area_Chair_V5iJ · 2024-12-19

**Metareview:**

The paper proposes a certification method for backdoor attacks in language models with random smoothing. The proposed method is a good attempt to develop a larger certification radius in the backdoor attacks, which is not widely studied in the robustness certification field. The experimental and theoretical results show the method is promising. The reviewers raise several valuable suggestions such as adding generation tasks other than classification and some potential baselines. Please include some in the main paper.

**Additional Comments On Reviewer Discussion:**

The author did a good job addressing all concerns raised by reviewers regarding baselines, settings, and tasks. Several interesting experiments have been added, which I highly encourage the author to include in the main paper.

---

### Decision · Program_Chairs · 2025-01-22

Accept (Poster)